# Lipocalin-2 (LCN2) Deficiency Leads to Cellular Changes in Highly Metastatic Human Prostate Cancer Cell Line PC-3

**DOI:** 10.3390/cells11020260

**Published:** 2022-01-13

**Authors:** Sarah K. Schröder, Manuela Pinoé-Schmidt, Ralf Weiskirchen

**Affiliations:** Institute of Molecular Pathobiochemistry, Experimental Gene Therapy and Clinical Chemistry (IFMPEGKC), RWTH University Hospital Aachen, D-52074 Aachen, Germany; saschroeder@ukaachen.de (S.K.S.); mpinoe@ukaachen.de (M.P.-S.)

**Keywords:** LCN2, NGAL, prostate cancer, CRISPR/Cas9, cytokines, adhesion, unfolded protein response (UPR), IL-1β, p-eIF2α, Cx43

## Abstract

The transporter protein lipocalin-2 (LCN2) also termed neutrophil-gelatinase-associated lipocalin (NGAL) has pleiotropic effects in tumorigenesis in various cancers. Since the precise role of LCN2 in prostate cancer (PCa) is poorly understood, we aimed to elucidate its functions in PCa in vitro. For this purpose, LCN2 was transiently suppressed or permanently depleted in human PC-3 cells using siRNA or CRISPR/Cas9-mediated knockout. Effects of LCN2 suppression on expression of different tumorigenic markers were investigated by Western blot analysis and RT-qPCR. LCN2 knockout cells were analyzed for cellular changes and their ability to cope endoplasmic stress compared to parenteral PC-3 cells. Reduced LCN2 was accompanied by decreased expression of IL-1β and Cx43. In PC-3 cells, LCN2 deficiency leads to reduced proliferation, diminished expression of pro-inflammatory cytokines, lower adhesion, and disrupted F-actin distribution. In addition, IL-1β expression strongly correlated with LCN2 levels. LCN2 knockout cells showed enhanced and sustained activation of unfolded protein response proteins when treated with tunicamycin or cultured under glucose deprivation. Interestingly, an inverse correlation between phosphorylation of eukaryotic initiation factor 2 α subunit (p-eIF2α) and LCN2 expression was observed suggesting that LCN2 triggers protein synthesis under stress conditions. The finding that LCN2 depletion leads to significant phenotypic and cellular changes in PC-3 cells adds LCN2 as a valuable target for the treatment of PCa.

## 1. Introduction

Despite improved treatment options established in the recent years, prostate cancer (PCa) remains one of the most common and deadly cancer types in men worldwide [1,2]. Since the 1980s, prostate-specific antigen (PSA) has been used as a standard diagnostic tool for PCa [1,3]. However, several studies highlighted the limitations in the use of PSA in diagnostics of PCa due to lack of cancer-specificity and false-positive testing [4,5]. Therefore, additional biomarkers are required for more accurate diagnostic of PCa.

Lipocalin-2 (LCN2) is a secreted 25-kDa-sized protein belonging to the lipocalin family. Members of this family are transporter proteins characterized by a conserved tertiary β-barrel structure, consisting of eight single-stranded antiparallel β-sheets [6,7]. LCN2 was originally isolated from neutrophil granulocytes and gained attention as an important player in the innate immune defense [8]. LCN2 contains a *N*-glycosylation site Asn^85^, which facilitates correct protein folding and function but is not required for secretion [9,10]. LCN2 influences not only physiological biological processes but also the development and spread of various types of cancer [11,12]. Several lines of evidence show that depending on the cell type and the tumor microenvironment (TME), LCN2 has pleiotropic effects and can either act as a tumor promoter or as a suppressor of tumorigenesis. A tumor suppressing role of LCN2 has been found for instance in colorectal [13] or pancreatic cancer [14], whereas LCN2 promotes tumorigenesis in breast [15] or prostate cancer [16]. Moreover, LCN2 is highly discussed as a prognostic and/or diagnostic marker in various types of cancer including PCa [12,17].

LCN2 positively correlates with clinicopathological characteristics in PCa patients [16,18,19]. The fact that LCN2 is secreted makes it particularly suitable as a potential non-invasive biomarker. Indeed, LCN2 has been detected in urine and serum of PCa patients [16,20,21,22,23,24]. Nevertheless, the underlying cellular and molecular mechanisms on how LCN2 affects PCa tumorigenesis are still poorly understood.

A common tool to study mechanisms of PCa progression are human cell lines, established from different metastatic sides of prostate carcinoma [25,26,27]. These in vitro tools also include the PCa cell lines LNCaP and PC-3. LNCaP cells originate from a metastatic lymph node of a PCa patient and show adherent growth behavior with an epithelial morphology [28]. PC-3 cells are derived from bone metastasis of a patient suffering from adenocarcinoma (grade IV) [29]. Compared to PC-3 cells, LNCaP cells are androgen-sensitive and express PSA [28]. In vitro as well as in vivo, PC-3 cells display higher metastatic potential compared to LNCaP cells [25]. For instance, in xenograft models, injection of PC-3 cells gained tumors that proliferate faster and are more invasive [30]. Importantly, there is a clear correlation of LCN2 expression and metastatic potential of different PCa cell lines [16]. Furthermore, LCN2 expression was linked to physiological or pharmacological induction of endoplasmic (ER) stress in human PCa cells [31].

Mechanistically, it is known that NF-κB, as well as the ERK/Slug axis, play a role in the regulation of LCN2 in PCa cells [16,22,23]. In addition, increased LCN2 levels lead to enhanced motility of PCa cells in soft agar, as well as an increase in epithelial-to-mesenchymal transition (EMT) markers [16,22]. More recently, it has been demonstrated that knockout of LCN2 correlates with an increased sensitivity to cisplatin-induced apoptosis [32].

To use LCN2 as a potential biomarker for PCa aggressiveness or as a target for therapeutic applications, a precise understanding of signaling pathways and functions is fundamental. Therefore, the aim of the present study was to investigate the cellular role of LCN2 by establishing a stable LCN2 knockout PCa cell line. Herein, CRISPR/Cas9 strategy was applied to eliminate LCN2 in the highly metastatic PC-3 cell line. We show that the newly generated LCN2-deficient PC-3 cells exhibited reduced proliferation, adhesion, increased inflammatory markers, and a disruption of F-actin stress fibers distribution. In addition, we demonstrate for the first time that the expression of IL-1β in PC-3 cells depends on the presence of LCN2. Furthermore, compared to parental PC-3 cells, LCN2-deficient PC-3 cells showed enhanced expression of unfolded protein response (UPR) markers and increased eIF2α phosphorylation in endoplasmic (ER) stress conditions. Our data suggest LCN2 as an important driver of the metastatic potential in PC-3 cells. Thus, LCN2 is potentially an interesting therapeutic target for the therapy of prostate cancers.

## 2. Materials and Methods

### 2.1. Cell Lines and Cultivation

The PCa cell lines PC-3 and LNCaP were used in this study [28,29]. PC-3 cells were obtained from the German office (LGC Standards GmbH, Wesel, Germany) of the American type culture collection (ATCC) (PC-3, CRL-1435). The cells were cultured in Dulbecco’s Modified Eagle Medium (DMEM) supplemented with 10% fetal bovine serum (FBS), 100 μg/mL streptomycin, 100 U/mL penicillin, 2 mM L-glutamine, and 1 mM sodium pyruvate (all from Sigma-Aldrich, Taufkirchen, Germany). LNCaP cells were kindly provided by Prof. Dr. Twan Lammers (Institute of Experimental Molecular Imaging (ExMI), RWTH Aachen University) and grown in RPMI 1604 medium (ThermoFisher Scientific, Waltham, MA, USA) containing 10% FBS and supplemented with 2.5 mM L-Glutamine, 100 U/mL penicillin and 100 μg/mL streptomycin. Both cell lines were cultured at 37 °C in a humid atmosphere with 5% CO_2_ and subcultured twice a week using Accutase^®^ (A6964, Sigma-Aldrich) detachment solution.

For experiments, cells were seeded in six-well plates or 10 cm dishes. Basal RNA and protein expression was isolated in cells that were grown for three days in standard cell culture medium containing 10% FBS.

### 2.2. Generation of Stable LCN2 Knockout Cell Lines

The CRISPR/Cas9 technology was chosen to generate stable LCN2 knockout PC-3 cell lines. A functionally validated human LCN2 knockout kit (OriGene Technologies, Herford, Germany, Cat# KN407685) was used, which consists of two gRNA vectors (gRNA1: ACAGGGCTAGGCCCAGCCAC, gRNA2: AGAGGGACCTTGCTCAGAGG) and the donor DNA containing LoxP-EF1A-tGFP-P2A-Puro-LoxP. The knockout cell lines were generated according to the manufacturer’s instructions with minor adaptions. In brief, PC-3 cells were seeded at a density of 4.5 × 10^5^ cells in six-well plates. At 80% confluency, co-transfection of gRNAs and donor DNA (each 1 µg) was performed using 4 µL Lipofectamine2000^®^ (#11668-019) in serum-free Opti-MEM (#31985-070, both from ThermoFisher Scientific).

Then, 48 h post transfection, the cells were split 1:10 and grown for additional three days and then split again. The cells were allowed to recover for four weeks before addition of puromycin (5 µg/mL, #A1113803, ThermoFisher Scientific). Thereafter, the medium was renewed every 2–3 days. In addition, cells were sorted for GFP expression by flow cytometry using BD FACSAria II. The resulting LCN2 deficient PC-3 cells were termed LCN2 knockout (KO) cells.

For isolation of single-cell colonies, array dilution method or cultivation in semi-solid medium was used. ClonaCell^TM^-TCS Medium (#03814, STEMCELL Technologies, Vancouver, BC, Canada) was purchased as a semi-solid medium and used according to the manufacturer’s recommendations. After approximately 20 days macroscopically visible colonies appeared. Subsequently, different single-cell-derived clones were picked and propagated in standard cell culture medium containing all supplements and 10% FBS. The different LCN2-KO cell lines established and used in this study were indexed by numbers (i.e., LCN2-KO#1, LCN2-KO#4, and LCN2-KO#8).

### 2.3. Treatments

Cells were seeded in six-well plates to 60–70% confluency and starved overnight in DMEM containing 0.5% FBS. LCN2-KO cells were treated with 100 ng/mL recombinant human LCN2 (rhLCN2, R&D Systems, #1757-LC) for 24 h before harvesting.

To analyze UPR mechanism in PC-3 and LCN2-KO cells, tunicamycin (TUN, Sigma-Aldrich, #T7765) or glucose deprivation conditions were used. TUN was dissolved in DMSO (10 mg/mL) and diluted to final concentration (0.1 or 1 µg/mL) in DMEM containing 0.2% FBS. To generate glucose deprivation conditions, the cells were starved overnight and then cultured in DMEM medium without glucose (ThermoFisher Scientific, #11966-025) with all additives and FBS (0.5% or 10%). Cells were incubated for 24 h or 48 h before harvesting and further analysis.

LCN2-KO cells were stimulated with conditioned medium (CM) from parenteral PC-3 cells. To obtain CM, PC-3, and LCN2-KO cells were seeded in six-well plates for 36 h in standard cell culture medium containing 10% FBS. As indicated, different ratios of CM (amount fresh to old medium) were used. In parallel, the LCN2-KO cells to be treated were seeded. After reaching 70% confluency, the cells were starved overnight and stimulated for 48 h with CM before harvesting for protein analysis.

### 2.4. Plasmid Transfection

For transfection of plasmid DNA, cells were seeded in six-well plates in standard cell culture medium. On the day of transfection, cells reached approximately 70% confluency. One hour prior transfection, the medium was replaced to standard cell culture medium without antibiotics. Then, the cells were transfected with 1.5 µg of pCMV-SPORT6-hLCN2 (clone IMAGp998B166030Q, BioScience LifeSciences, Nottingham, UK) encoding full-length human LCN2 using Metafectene^®^ (#T020-1.0, Biontex, Munich, Germany) as transfection reagent according to the manufacturer’s instructions. As a control, the cells were transfected with an empty vector (EV) construct (pCMV-SPORT6). Then, 24 h or 48 h after transfection, the cells were harvested for protein analysis. In case cells were stimulated after transfection, medium was changed to starving medium (0.5% FBS) 6 h post transfection. Stimulations were performed the next day, as described above.

### 2.5. Adenoviral Constructs and Infection

Adenoviral infection was used as to transiently re-express LCN2 in LCN2-deficient cells. Ad5-CMV-hLCN2 was cloned following a similar strategy, as described for Ad5-CMV-mLCN2 [33]. In brief, the vector pCMV-SPORT6-hLCN2 encoding full-length human LCN2 obtained from Source BioSciences Life Sciences, Nottingham, UK (clone IMAGp998B166030Q) was digested with *Sal*I and *Hind*III and the resulting ~900 bp fragment was cloned in the adenoviral shuttle vector pShuttle-CMV (Stratagene, Agilent, La Jolla, CA, USA). The integrity of the resulting vector pShuttle-CMV-hLCN2 was confirmed by restriction digest and sequence analysis. The shuttle vector was linearized by cutting with restriction enzyme *Pme*I, dephosphorylated, and transferred into the adenoviral backbone vector pAd-Easy-1 (Stratagene). One recombinant clone (AdEasy-CMV-hLCN2) was validated by restriction digest and sequence analysis. Thereafter recombinant viral particles were produced and purified by CsCl gradient centrifugation and gel filtration, as described before [33]. Infections with adenoviral vector Ad5-CMV-GFP were performed in parallel to control cellular infection. For infection, the cells were seeded on 10 cm dishes to reach 80% confluency in DMEM (with all supplements) containing 10% FBS. The cells were infected for 6 h with 2 × 10^8^ pfu/mL Ad5-CMV-hLCN2 or Ad5-CMV-GFP in DMEM (all supplements) containing 0.5% FBS. After changing medium to DMEM (all supplements) containing 10% FBS, infected cells were cultured for additional 48 h before harvesting for protein extraction.

### 2.6. siRNA-Mediated Silencing of LCN2

LCN2 was transiently silenced in PC-3 cells using the validated siRNA HS-LCN2-6-HP (#SI03246908, Qiagen, Hilden, Germany) [34,35,36]. As a non-silencing control, AllStars negative control siRNA AF 488 (#1027284, Qiagen), which is validated to have no homology to any known mammalian gene, was used. For siRNA-delivery, PC-3 cells were seeded in six-well plates in standard cell culture medium. On the day of transfection (at least 60% confluency), the medium was changed to standard culture medium without antibiotics 1 h prior transfection. Then, 5 to 35 nM siRNA against LCN2 (siLCN2) was delivered into PC-3 cells using 5 μL Lipofectamine^®^ 2000 transfection reagent in serum-free Opti-MEM according to the manufacturer’s instructions. Silencing of LCN2 mRNA and protein expression was monitored 48 h post transfection. When the cells were further stimulated after siRNA-mediated knockdown, they were starved overnight before stimulation on the next day.

### 2.7. Quantitative mRNA Analysis

Quantitative mRNA analysis was performed as previously described [37]. In brief, total RNA was isolated and purified. cDNA was synthetized from 1 µg RNA using SuperScript II reverse transcriptase (#18064-022, ThermoFisher Scientific).

For RT-qPCR, primers were purchased from Eurofins Genomics Germany GmbH (Ebersberg, Germany). Primer sequences were either designed using the Universal ProbeLibrary offered by Roche Diagnostics (Mannheim, Germany), using the NCBI Primer Blast tool, or alternatively synthesized according to published sequences (Appendix A). Whenever possible, human primers were positioned intron spanning.

At least three independent experiments were carried out in the individual test series. One experiment was performed in technical triplicates measured as duplicates and normalized to either glyceraldehyde 3-phosphate dehydrogenase (*GAPDH*) or beta actin (*ACTB*) expression. Relative mRNA expression was calculated using the 2^−ΔΔCT^ method [38] and expression levels were represented as the normalized quantity of target mRNA relative to the normalized quantity of control mRNA.

### 2.8. Protein Analysis

For analysis of secreted proteins, supernatants were collected, centrifuged (4 °C, 10,000 rpm, 10 min) to remove detached cells, and stored at −20°C for further use. For total cell lysates, cells were harvested in RIPA buffer (50 mM Tris-HCl (pH 7.2), 150 nM NaCl, 1% (*w*/*v*) NP-40, 0.1% (*w*/*v*) SDS, 0.5% (*w*/*v*) sodium deoxycholate) containing Complete™—mixture of proteinase inhibitors (#11849300, Roche Diagnostics) and phosphatase inhibitor cocktail II (#P5726, Sigma-Aldrich, Taufkirchen, Germany). The protein content of each sample was determined by the DC protein assay (#500-0116, Bio-Rad Laboratories GmbH, Düsseldorf, Germany). Equal amounts of proteins (45 µL of supernatants and 20–40 µg for total cell lysate) were mixed with dithiothreitol (DTT) as a reducing agent and Nu-PAGE™ LDS electrophoresis sample buffer (#NP0008, ThermoFisher Scientific). Before separating proteins in 4–12% Bis-Tris gradient gels using 2-(N-morpholino)ethanesulfonic acid (MES) running buffer, samples were heated at 80 C for 10 min for denaturation. Proteins were electroblotted on nitrocellulose membrane (#GE10600002, 0.45 μm, Merck, Darmstadt, Germany) and equal protein loading and successful transfer was confirmed by Ponceau S stain. Non-specific binding sites were blocked with 5% (*w*/*v*) non-fat milk powder in Tris-buffered saline with Tween 20 (TBST). Membranes were incubated with primary antibodies overnight (shaking, 4 °C) and subsequently visualized with secondary antibodies coupled to horseradish peroxidase with SuperSignal chemiluminescence substrate (#34076, ThermoFisher Scientific). The details of all primary and secondary antibodies used in this study are listed in Appendix A.

### 2.9. Thiazolyl Blue Tetrazolium Bromide (MTT) Assay

A colorimetric thiazolyl blue tetrazolium bromide (MTT) assay was performed to determine metabolic activity of parenteral PC-3 and LCN2-KO cells. 10,000 cells/well were seeded in 96-well plates in standard cell culture medium. After incubation for 24 h or 48 h, 10 μL of a sterile MTT stock solution (#ab146345, Abcam, Cambridge, UK) prepared in phosphate-buffer saline (PBS) at 5 mg/mL were added to each well. After 4 h, 100 μL DMSO containing 0.04 N HCl were added to each well to solubilize formazan crystals. The optical density was determined with Victor X3 multilabel plate reader (PerkinElmer, Waltham, MA, USA) at an absorbance of 570 nm with a reference at 690 nm.

### 2.10. Cell Adhesion Assay

Parenteral PC-3 and LCN2-KO cells were used for an indirect cell attachment assay, as it has been previously described by Hu and coworkers [39]. For this purpose, all cell lines were seeded in a density of 3 × 10^5^ cells in 12-well plates. The cells were cultured for 5 h at 37 °C to allow cell attachment. Thereafter, the cell culture medium was carefully collected for each well. The cells were washed once with PBS, and this was mixed with the collected cell culture medium. Non-adherent cells from each well were counted under a light microscope in a hemocytometer. Mean values of three independent experiments (each in triplicates) were determined and values expressed as a percentage of the total seeded cells.

### 2.11. Phalloidin Staining

Parenteral PC-3 and LCN2-KO cells were seeded and cultured for 24 h on coverslips at a density of ~50,000 cells/well in 24-well plates. After washing with PBS, cells were fixed with 3.7% paraformaldehyde at room temperature for 20 min. Afterwards, the cells were washed again with PBS before permeabilization in precooled 0.1% sodium citrate/ 0.1% Triton-X-100 solution for 3 min on ice. All following steps were performed at room temperature and exclusion of light. Subsequently, cells were washed with PBS and 50% FBS/0.5% bovine serum albumin (BSA) solution (in PBS) for 1 h on a shaker. The Phalloidin-Rhodamine solution (#R415, ThermoFisher Scientific, 40×) was diluted to 1× in PBS containing 1% BSA. The cells were stained with Phalloidin-Rhodamine for 20 min and counterstaining of the nuclei was performed with DAPI (#D1306, ThermoFisher Scientific, 200 ng/mL) for 30 min. After washing steps with PBS and H_2_O, the cells were mounted with PermaFluor aqueous mounting medium (#TA-030-FM, ThermoFisher Scientific) and images of the cytoskeleton were acquired using a Nikon Eclipse E80i fluorescence microscope (Nikon Imaging Japan Inc., Shinagawa-ku, Tokyo, Japan).

### 2.12. Data Analysis

Unless otherwise stated, the results derived from at least three independent experiments were expressed as the means of the group plus standard derivation, calculated with Microsoft Excel (Microsoft Corporation). For statistical analysis, GraphPad Prism (GraphPad Software version 6) was used.

Gaussian distribution was examined using the D’Agostino and Pearson omnibus normality test. In case of normal distribution, unpaired Student’s *t*-test and otherwise non-parametric Mann–Whitney U test was performed. For comparison of means of more than two groups, one-way analysis of variances (ANOVA) with Tukey multiple comparisons test, or Kruskal–Wallis test with Dunn’s multiple comparisons test were used for parametric or non-parametric distributions, respectively.

Two-way ANOVA used to examine how genotype and treatment determine response to TUN-induced ER stress in PC-3 and LCN2-KO#1 cells. Tukey’s test was applied as post hoc test. Probability values of *p* < 0.05 were considered significant. Differences between the group means reaching significance are indicated as * *p* < 0.05, ** *p* < 0.01, and *** *p* < 0.001, respectively.

## 3. Results

### 3.1. LNCaP and PC-3 Cells Differently Express Tumorigenic Markers

Reports in recent decades provided evidence that LCN2 expression unfavorably correlates with PCa progression in vitro and in xenograft models [16,18,19].

We confirmed a correlation between LCN2 expression and pro-metastatic markers in the human PCa cell lines LNCaP and PC-3 by Western blot analysis and RT-qPCR. Compared with the lower metastatic LNCaP cells, the highly metastatic PC-3 cells expressed approximately 150-fold more LCN2 mRNA (Figure 1A) and showed elevated more LCN2 protein expression (Figure 1B) than LNCaP cells. The higher expression of LCN2 in PC-3 cells was accompanied by increased mRNA and protein expression of *IL1B*, *NFKBIZ,* and *CXCL8*. Furthermore, gap junction protein α-1 (*GJA1*) mRNA encoding connexin-43 (Cx43), which is important for cellular adhesion and communication [40], was around 5000-fold more abundant in PC-3 cells.

### 3.2. Effects of siRNA-Mediated Knockdown of LCN2 in PC-3 Cells

To investigate the function of LCN2 in PCa, siRNA-mediated knockout technique was performed. For this purpose, due to its high metastatic potential and the high LCN2 expression, the PC-3 cell line was chosen. PC-3 cells were transfected with different concentrations of siRNA (siLCN2), and LCN2 protein expression and secretion was examined post transfection (Appendix A). Potent, transient downregulation of LCN2 in PC-3 cells was achieved with 35 nM siLCN2 (Figure 2A) after 48 h. Compared with untreated PC-3 cells, the use of scrambled (scr) control siRNA showed no effect on LCN2 or GAPDH expression. Similarly, RT-qPCR confirmed the suppression of LCN2 protein expression (~73%) using 35 nM siLCN2 compared with scr siRNA control (Figure 2B).

Knockdown of LCN2 is known to affect tumorigenic markers and the EMT process in various cancers [12,17]. Our results show decreased protein expression of fibronectin, connexin 43 (Cx43) and nuclear factor kappa inhibitor zeta (IκBζ) when LCN2 was transiently downregulated with 35 nM siLCN2 in PC-3 cells (Figure 2C). Furthermore, we observed a decrease in IL-1β expression 48 h post transfection in the siLCN2-transfected PC-3 cells.

Taken together, the transient downregulation of LCN2 unraveled novel downstream targets of LCN2 in PC-3 cells. Next we investigated whether these are also relevant in a stable LCN2-deficient cell line.

### 3.3. Generation of LCN2-Deficient PC-3 via CRISPR/Cas9 Technology

In contrast to transient knockdown techniques, the permanent knockout of a specific gene results in full depletion of the functional protein [41]. To study the function of LCN2 in metastatic PCa, we established a LCN2-deficient PC-3 cell lines using the CRISPR/Cas9 technology. Details of the procedure and generation of these cells are displayed in the Materials and Methods section. After puromycin selection, a stable pool of transfected PC-3 cells was established. Western blot analysis confirmed successful transfection procedure as the transfected PC-3 cells showed turbo GFP expression (Appendix A). In addition, compared with the parenteral PC-3 cells, there was no visible LCN2 protein expression in the transfected cells.

Subsequently, individual single-cell-derived PC-3 LCN2-knockout (KO) cell lines were established using the semi-solid medium method. After 20 days of culturing, individual single-cell derived colonies were visible (Figure 3A). Microscopical images verified that all cells within these colonies were positive for GFP (Figure 3B). The generated different colonies were transferred to normal liquid standard cell cultured medium and traded as individual PC-3 LCN2-KO cell lines, indicated by different numbers (e.g., LCN2-KO #4). Western blot analysis confirmed full depletion of LCN2 protein expression and secretion in all LCN2-KO cell lines (Figure 3C). In further experiments, the LCN2-KO cell lines #1, #4, or #8 were used. All analyzed LCN2-KO cell lines showed a complete loss of *LCN2* mRNA expression compared to parenteral PC-3 cells (Figure 3D).

After successful generation of different individual PC-3 LCN2-KO cell lines, the aim was to analyze phenotypic and cellular changes in these cell lines resulting from LCN2 depletion.

### 3.4. LCN2-Deficient PC-3 Cells Show Reduced Proliferation, Adhesion, and Disrupted F-Actin Stress Fibers

LCN2 can either promote or suppress tumorigenesis in different cancer types [42,43]. In PCa, data indicate a positive correlation between LCN2 and metastatic spread, including effects on EMT process and proliferation [16,23,32,44]. A previous report demonstrated that the overexpression of LCN2 promotes cell migration and invasion via ERK activation and SLUG expression [16]. Therefore, we first comparatively analyzed *SNAI1*, *SNAI2*/*SLUG,* and *TWIST* mRNA expression in LCN2-KO cells and parenteral PC-3 cells. We found that only *SNAI2*/*SLUG* was slightly decreased in LCN2-KO#1 cells (not shown), suggesting that this pathway was not significantly altered in respective cells.

Routine splitting of the cells showed that the LCN2-KO cell lines grew more slowly than the parenteral PC-3 cells. Reduced proliferation in LCN2-KO cell lines was reflected by significantly decreased expression of proliferating cell nuclear antigen (*PCNA*) expression in RT-qPCR analysis (Figure 4A). Western blot analysis confirmed reduced protein expression of PCNA in all analyzed LCN2-KO cell lines when compared to PC-3 cells (Figure 4B). Furthermore, in comparison to PC-3 cells, LCN2-KO cell lines showed significantly reduced metabolic activity after 48 h of cultivation as determined by MTT assay (Figure 4C).

Metastatic spread of tumor cells is a highly complex process including cell–cell communication and cellular adhesion [45]. It is known that *GJA1*/Cx43 is involved in this process and correlated with malignancy in PCa cells [46]. We observed significantly reduced *GJA1* mRNA expression in analyzed LCN2-deficient cell lines compared to PC-3 cells (Figure 4A). Similarly, LCN2-KO cell lines exhibited markedly decreased Cx43 protein levels (Figure 4B). In addition, data analysis by one-way ANOVA from indirect cell attachment assay showed significant differences between PC-3 and LCN2-KO cell lines (Figure 4D). Whereas almost all PC-3 cells (94.6% ± 0.8) were able to attach within 5 h after seeding, only around half of the LCN2-KO#1 cells (46.6% ± 8.7) were attached. Similarly, the other examined LCN2-KO cell lines showed with 65.8% ± 1.4 (LCN2-KO#4) and 57.4% ± 4.6 (LCN2-KO#8) a significant reduction in cell adhesion (Figure 4E).

F-actin stress fibers are critical cellular structures that contribute to adhesion and cytoskeleton rearrangement [47]. To visualize F-actin stress fibers in our cell lines, we performed Phalloidin-Rhodamine staining with nuclear counterstaining by DAPI. PC-3 cells displayed an organized F-actin stress fiber distribution (Figure 4F).

Contrarily, LCN2-deficient cells showed a disrupted F-actin stress fibers appearance. This was evident in LCN2-KO#1 cells by a disordered, more rounded, condensed shape of F-actin stress fibers. These changes were as well observed in the LCN2-KO#4 and LCN2-KO#8 cells (Appendix A), suggesting LCN2 significantly affects metastatic properties of PC-3 cells such as proliferation and adhesion. In particular, the expression of the adhesion protein Cx43 appears to be regulated by LCN2.

### 3.5. IL-1β Expression Correlates with Presence of LCN2 in PC-3 Cells

In some cancers, tumor progression is strongly affected by expression of pro-inflammatory cytokines in the TME [48]. In PCa, inflammation is discussed as a key driver or precursor of advanced cancers [49]. Since the LCN2-KO cells generated in this study have lower metastatic characteristics, we speculate that likewise the expression of pro-inflammatory cytokines may be decreased.

Indeed, RT-qPCR and Western blot analysis revealed a significantly decreased expression of pro-inflammatory cytokines and associated mediators in all LCN2-KO cell lines compared to parenteral PC-3 cells (Figure 5A,B). In detail, we observed a strong reduction in *NFKBIZ*/IκBζ, *IL1B*/IL-1β and *CXCL8*/IL-8 in all analyzed LCN2-deficient PC-3 cells at mRNA and protein level.

In both primary hepatocytes and cancer cell lines, IL-1β is known as a potent inducer of LCN2 [9,50]. However, human PC-3 cells do not enhance LCN2 expression when stimulated with IL-1β [37]. The results of the aforementioned experiments suggest that LCN2 might regulate IL-1β levels in PC-3 cells. To clarify this, LCN2 was added as rhLCN2 in the medium or restored in LCN2-deficient cells by different methods. In one set of experiments, LCN2-KO#1 cells were stimulated for 48 h with CM collected from either PC-3 or LCN2-KO#1 cells to observe whether medium from LCN2-expressing cells can affect IL-1β expression. Western blot analysis revealed that in both concentrations/ratios used (50% or 100%), CM derived from PC-3 cells induced IL-1β (Figure 5C). In contrast, treatment with fresh medium compared with CM of LCN2-KO#1 cells in LCN2-KO#1 cells showed only a slight increase in IL-1β levels. Similarly, the direct stimulation with 100 ng/mL rhLCN2 led to a slight increased IL-1β expression in LCN2-KO#1 cells (Figure 5D). In a second set of experiments, LCN2 was restored in LCN2-KO#1 cells by transient transfection (Figure 5E). Similarly to the stimulation with recombinant protein, LCN2 reconstitution was accompanied by increased IL-1β production in LCN2-KO#1 cells when compared to EV-transfected control cells. Moreover, the restoration of LCN2 expression by adenoviral infection already at a low infection rate was able to enhance IL-1β expression, while the reporter virus Ad5-CMV-GFP directing expression of GFP also under transcriptional control of the CMV promoter failed to impact IL-1β expression in LCN2-KO#1 cells (Figure 5F). In contrast, changes of IL-1β protein expression were undetectable in LNCaP cells when supplemented with LCN2 (not shown). This might be due to the fact that IL-1β protein expression is rather low under basal conditions (cf. Figure 1B).

Since different methods of restoring LCN2 or adding LCN2 show very similar findings, we concluded that LCN2 expression affects the level of IL-1β in PC-3 cells, which is important for advanced PCa.

### 3.6. LCN2-Deficient PC-3 Cells Are Prone to Endoplasmic Reticulum Stress and Unfolded Protein Response

Cancer cells maintain survival under a wide range of stressful environmental conditions, such as cytokine excess, nutrient deprivation, or ER stress [51]. They evoke different mechanism to cope with these challenging conditions [52]. There is evidence that LCN2 has protective functions in different harmful conditions in vitro and in vivo [35,53]. However, it is uncertain whether the presence of LCN2 in PC-3 cells affects cell behavior in stressful environmental conditions. Therefore, LCN2-KO#1 and PC-3 cells were exposed to different ER stress-inducing conditions and UPR marker were investigated.

TUN is a natural occurring antibiotic triggering ER stress through prevention of *N*-glycosylation [54,55]. LCN2 has an evolutionarily conserved *N*-glycosylation side that is linked to biantennary complex glycans with or without outer arm fructose and sialic acid [9,53].

When PC-3 cells were stimulated for 24 h with TUN, Western blot analysis confirmed accumulation of non-glycosylated LCN2 protein expression (Figure 6A). This shift from around 25-kDa-sized LCN2 to a smaller non-glycosylated from (*LCN2) was visible in both cellular and secreted protein fractions. Moreover, the stimulation with TUN resulted in an activation of different UPR markers, namely BIP, p-eIF-2α, ATF-4 and CHOP (Figure 6B). We have previously shown that NF-κB signaling is relevant for LCN2 expression during UPR in primary hepatocytes [53]. In line with these findings, we observed increased NF-κB activation in LCN2-KO#1 cells during ER stress (not shown). However, the strength of induction was different in some UPR markers between LCN2-KO#1 and PC-3 cells. Whereas LCN2-KO#1 cells showed strong increase in ATF-4 and p-eIF2α (Ser51) protein levels when treated with TUN, PC-3 cells displayed only slight induction of these markers. RT-qPCR analysis revealed that both PC-3 and LCN2-KO#1 significantly induced *ATF4* mRNA expression when treated with 1 µg/mL TUN. *LCN2* expression was not significantly affected in TUN-induced ER stress conditions. Both cell lines significantly diminished proliferation (*PCNA* expression) in a similar manner when treated with TUN (Figure 6C). Furthermore, we found that TUN-induced ER stress lasted longer in LCN2-KO#1 than in PC-3 cells, as different UPR markers are still strongly expressed after 72 h. However, neither cell line showed signs of apoptosis under sustained UPR (Appendix A).

Based on these results, we proposed that the presence of LCN2 might affect phosphorylation of p-eIF2α because even untreated conditions showed higher levels of p-eIF2α in LCN2-KO#1 cells compared to PC-3 cells. To test this hypothesis, transiently LCN2-overexpressing LCN2-KO#1 cells and siLCN2-transfected PC-3 cells were generated. 24 h post transfection, cells were treated with TUN (1 µg/mL) for additional 24 h and phosphorylation of eIF2α was monitored on protein level (Figure 6D). The siRNA-mediated knockdown of LCN2 followed by TUN treatment resulted in an enhanced occurrence of p-eIF2α in PC-3 cells. A reverse correlation was found in LCN2-KO#1 cells as the quantities of p-eIF2α were strongly reduced in the LCN2-overexpressing cells in relation to the cells transfected with EV. In addition, it was shown that proper glycosylation of successfully restored LCN2 in LCN2-KO#1 cells was prevented by TUN treatment as there was a strong shift from LCN2 to *LCN2.

We finally investigated whether the observed effects are limited to TUN or can be as well seen under other ER stress-inducing conditions. To do so, PC-3 and LCN2-KO#1 cells were grown in normal growth medium containing glucose and without glucose for 48 h including different amounts of FBS (0.5% or 10%). In PC-3 cells cultured under glucose deprivation conditions, strong accumulation of non-glycosylated LCN2* was observed, indicating ER stress in the cells (Figure 6E). This was confirmed in both cell lines by activation of ATF-4 and p-eIF2α. As seen for TUN, culturing without glucose led to a stronger activation of UPR markers, especially p-eIF2α in LCN2-KO#1 cells compared with parenteral PC-3 cells. The different serum concentrations had no impact on the seen effects, but there was an increased UPR visible with lower serum concentration.

In summary, we found that knockdown or depletion of LCN2 in PC-3 cells results in increased UPR upon TUN treatment or glucose deprivation. Particularly, p-eIF2α activation is an important response in LCN2-KO#1 cells to possible cope ER stress. All findings provide strong evidence that LCN2 plays a critical role in stressful conditions in PCa cells.

## 4. Discussion

Studies of the past decades revealed that LCN2 plays an essential role in tumorigenesis of various cancers. Initial screening of publicly available data already suggests that LCN2 expression is significantly altered during initiation and progression of PCa (Appendix A). Presently, simple somatic mutations and copy number variations of LCN2 are more frequent described in other malignancies than PCa (Appendix A). Although there is already discussion whether LCN2 can be used as a reliable biomarker [17,19], there is still a lack of understanding on its exact function in tumor progression, especially in PCa.

Herein, we confirmed that LCN2 expression in PCa cell lines correlates with expression of other metastatic markers. Compared to the LNCaP cells, the PC-3 cells showed about 150-fold increased *LCN2* mRNA expression and higher LCN2 protein. This is in line with previous studies published by Ding and colleague [16]. However, in the previous study, only a 4-fold relative increase in *LCN2* mRNA expression between LNCaP and PC-3 cells was reported [16]. The differences in fold-change might be explained by the different normalization genes chosen. Furthermore, this may be due to differences in cultivation conditions as well as time since it is known that the detectable amount of LCN2 increases during prolonged culturing [50].

Our studies showed that Cx43, an important molecule in cellular adhesion and cell-cell communication [40], is much more strongly expressed in the highly metastatic PC-3 cells than in the LNCaP cells. As pro-inflammatory cytokines and chemokines contribute to the development as well as the spread of PCa [49,56], it was expected that the cell lines express different levels of inflammatory mediators. Similarly, we found that the highly metastatic PC-3 cells show significantly higher mRNA and protein expression of *CXCL8, NFKBIZ,* and *IL1B* when compared to LNCaP cells. The initial comparison between the PCa cell lines PC-3 and LNCaP confirmed that various tumor-associated markers also correlate with metastatic potential in vitro. The use of LNCaP cells as a model for early-stage metastasis and PC-3 cells for late, aggressive stage is well accepted. However, the two cell lines were originally established from metastasis in different organs, which means that factors in the TME from which they originate (lymph node: LNCaP; bone marrow: PC-3) might differentially imprinted them. In addition, a comprehensive genetic analysis by Dozmorov and coworkers showed that although the two cell lines show some common biochemical features, they display a variety of unique signaling pathways [57]. Therefore, findings established in LNCaP and PC-3 cells are only comparable to a limited extent when correlating LCN2 with other tumorigenic markers. Nevertheless, the data from the cell lines provide initial evidence that LCN2 affects cell-cell communication and pro-inflammatory signaling in PCa.

To investigate the influence of LCN2 more precisely, we transiently suppressed LCN2 in PC-3 cells using targeted siRNAs and further established LCN2-deficient cell lines. Both approaches results in similar findings. We demonstrated a positive correlation of LCN2 on the adhesion ability in PCa cells. An indirect cell attachment assay revealed significantly lower adhesion of all LCN2-deficient PC-3 cell lines compared with their parenteral PC-3 cells. Similarly, Hu and coworkers found that LCN2 mediates attachment to basement membrane in colon cancer. Compared to KM12C colon carcinoma cells in which LCN2 was downregulated, LCN2-overexpressing cells showed a significant increase in adhesion [39]. Similarly, LCN2 improves adhesion of pancreatic ductal adenocarcinoma cells on collagen I and fibronectin substrata [58].

Since LCN2 is known to promote PCa migration [18,32], we wanted to know whether cytoskeletal organization is also affected by the persistent LCN2 knockout in PC-3 cells. Staining of F-actin stress fibers by Phalloidin revealed a disrupted organization of the cytoskeleton in LCN2-KO cells. Whereas PC-3 cells are described as migrating cells that contain lamellipodia and filopodia [29], the newly generated LCN2-deficient PC-3 cell lines showed cellular characteristics favoring reduced migration capacity. Conversely, the loss of adhesion is typically associated with altered phenotype and metastatic spread [59]. This discrepancy might be due to the fact that tumorigenesis is a complex interplay of many steps, including detachment of the primary tumor, followed by increased invasion and adhesion to form metastases. Similar observations have been made in ovarian cancer cells, where increased adhesion, migration, and invasion contribute to metastasis [60]. Mechanistically, it is possible that MEK/ERK signaling axis is another critical factor that contributes to altered F-actin rearrangement as it was previously shown in esophageal squamous cell carcinoma cells [61]. The reduced expression of the proliferation marker PCNA, as well as the reduced metabolic activity of the cells, also suggests a lower metastatic potential of LCN2-KO cells.

In terms of adhesion, cell–cell communication is also important for cancer cells. Studies by Zhang and colleagues provide evidence that Cx43 is related to malignancy in PCa [46]. Herein, we describe, to our knowledge for the first time, interplay between LCN2 and Cx43 in PC-3. On the one hand, this was shown by a diminished expression of Cx34 protein when LCN2 was temporarily repressed by siRNA in PC-3 cells. In addition, different individual LCN2-deficent PC-3 cell lines exhibit markedly decreased Cx43 expression and significantly lower *GJA1* mRNA levels compared to parenteral PC-3 cells. Further studies are now needed to investigate more precisely the molecular signaling pathways through which LCN2 and Cx43 interact. A conceivable molecule that interplays with both Cx43 and LCN2 might be the focal adhesion kinase (FAK) [62,63]. However, this hypothesis needs to be tested.

Furthermore, we found that a decrease in LCN2 is associated with a reduced expression of several inflammatory markers, which contribute to malignancy in PCa [64,65,66]. Both, using siRNA targeting LCN2 and LCN2-deficient cell lines generated by CRISPR/Cas9 showed decreased expression of IL-1β, IκBζ, and IL-8. Although it is known from endometrial cancer that LCN2 can mediate the expression of cytokines [67], there are hardly any studies regarding PCa. Interestingly, it was recently demonstrated that chronic administration of IL-1α or IL-1β changed the lower metastatic LNCaP cells to a more aggressive, malignant cancer type [66]. This, and a correlation of IL-1β with the Gleason score, make it an interesting target in PCa [68]. Possibly PCa cells varying in responsiveness towards estrogen or androgen might react differently regarding LCN2 stimulation and IL-1β expression/induction. In particular the two cell lines analyzed here (i.e., PC-3 and LNCaP) are known to differ in androgen sensitivity [30]. Therefore, regulatory networks existing in one PCa cell line do not necessarily have to be generally valid.

We further investigated whether IL-1β expression is dependent on presence of LCN2. Restoration of LCN2 via transfection or infection, as well as addition of rhLCN2 in LCN2-KO#1 cells, showed the presumed upregulation of IL-1β. In a previous study, we have already demonstrated that cytokines regulate LCN2 in PCa cells [37], whereas based on the current data, we provide the first evidence that LCN2 also affects cytokine expression, particularly IL-1β, in PCa. However, the pathways by which LCN2 trigger IL-1β expression needs to be investigated.

In the last part of our study, we assessed the effects of LCN2 in relation to ER stress. Both the use of TUN, which prevents *N*-glycosylation and thus leading to an accumulation of misfolded proteins [54], and the deprivation of glucose for 48 h resulted in activation of several UPR markers in both PC-3 and LCN2-KO#1 cells. We observed no increase in LCN2 expression in TUN-treated PC-3 cells or PC-3 cells upon glucose deprivation. Rather, we could not only prove the prevention of *N*-glycosylation of LCN2 by TUN, as previously reported [10], but also that presence of glucose is essential for correct *N*-glycosylation of LCN2. Interestingly, glucose starvation in cancer cells can lead to activation of cell death via UPR [69]. In the present study, neither PC-3 nor LCN2-KO#1 showed signs of apoptosis and may have evolved other mechanism to cope with these stress factors. However, we observed in both cell lines decreased proliferation when treated with TUN. This cell cycle arrest has been already described elsewhere [70].

Some other studies have already described a protective role for LCN2 under diverse harmful stress conditions [34,71,72], thus we propose that LCN2 might have a protective role in PCa as well. In line with this assumption, LCN2-KO#1 cells showed much stronger and prolonged activation of UPR markers compared to parenteral PC-3 cells. This was shown in basal condition, as well as when ER stress was induced by TUN or glucose deprivation. More precisely, present findings provide evidence that LCN2 deficiency leads to dramatically increase in phosphorylation of p-eIF2α in PCa cells. We speculate that this activation of p-eIF2α represents a compensatory protective effect in LCN2-KO#1 to handle ER stress. This assumption is supported previous studies, as the activation of p-eIF2α is a fast mechanism in mammals to control stress responses and to counteract the accumulation of unfolded proteins [73,74]. Furthermore, p-eIF2α controls survival under glucose starvation [75]. Presently, the mechanism by which LCN2 exerts a protective effect on the PC-3 cells can only be assumed. Our findings let us speculate that NF-κB is involved, as NF-κB signaling is stronger elevated in LCN2-KO#1 cells compared to PC-3 cells under stressful conditions. This is in line with previous findings, which show that primary hepatocytes and bone marrow-derived macrophages isolated from Lcn2-deficient mice are more susceptible to lipopolysaccharide treatment, in which NF-κB signaling plays a key role in mediating induction of various pro-inflammatory responses [53,76]. Subsequent studies, based on these findings could show whether these activated mechanisms also result in altered behavior with respect to chemotherapeutic agents.

It is well known that LCN2 is a pleiotropic mediator that is regulated by many different pathways. In addition, LCN2 effects are influenced by sex-specific factors [77,78,79]. Similar to breast cancer [80], we speculate that estrogen receptor alpha (ERα) might also be related to LCN2 in PCa. However, there is conflicting data regarding the interaction between LCN2 and ERα in general, in different organs, and different genders [76]. Regarding invasion, migration and colony formation, others have recently shown that estrogen receptors play a promoting role in PC-3 cells [79]. In addition, the promoter region of *LCN2* contains an estrogen response element [77], which might result in a direct dependency of LCN2 expression and estrogen content. Furthermore, the stimulation of PC-3 cells with estrogen was shown to activate ERK1/2 [81], possibly indicating that the activation of ERK1/2 and LCN2 might be indirectly correlated. Studies investigating the impact of estrogen signaling in the above discussed pathways for the pathogenesis of PCa are urgently needed.

## 5. Conclusions

In conclusion, the results of the present study highlight LCN2 as a valuable target in advanced PCa, as a LCN2 deficiency greatly diminished the metastatic characteristics of PC-3 cells. Furthermore, depletion of LCN2 identified further downstream targets of LCN2 in PC-3 cells. Now it will be of fundamental interest to test if the in vitro findings are transferrable to PCa patients. It is important to know whether there is a correlation between high LCN2 and inflammatory response, as both strongly contribute to advanced PCa. Blocking these components or signaling pathways could open new therapeutic approaches for more effective PCa therapies. In future, it will be necessary to conduct studies investigating more detailed mechanistic aspects explaining the molecular involvement of LCN2 in pathways associated with the pathogenesis of PCa. In particular, animal experiment, such as an orthotopic PCa mouse model is required to validate the involvement of LCN2 in PCa and metastasis.

## Figures and Tables

**Figure 1 cells-11-00260-f001:**
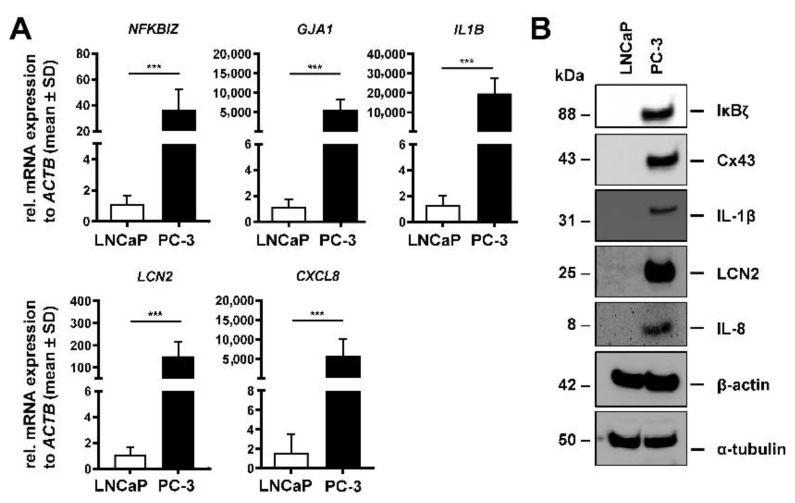
LCN2 and other tumorigenic markers in human prostate cancer cell lines. The human prostate cancer cell lines LNCaP and PC-3 were analyzed for mRNA or protein expression after culturing under basal conditions. (**A**) Quantitative mRNA analysis of *NFKBIZ, GJA1, IL1B, LCN2,* and *CXCL8* in both cell lines revealed significantly higher expression levels in PC-3 cells (n = 4). Relative mRNA expressions were normalized to *ACTB* expression and values are given in relation to LNCaP cells. Significant differences between LNCaP and PC-3 cells are indicated as *** *p* < 0.001. (**B**) Western blot analysis confirmed lower protein expression of these markers in LNCaP cells (n ≥ 3). β-actin and α-tubulin expression served as internal loading controls.

**Figure 2 cells-11-00260-f002:**
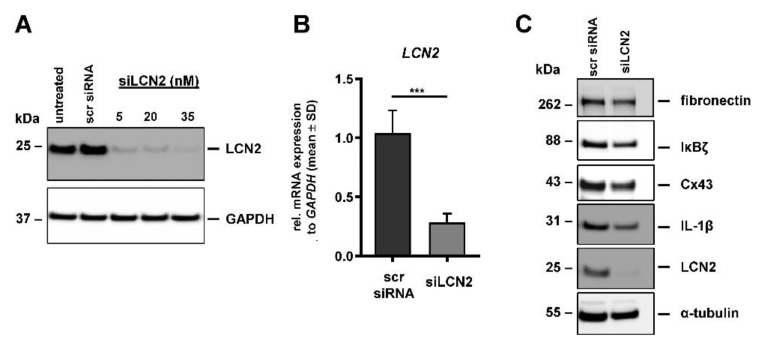
siRNA-mediated knockdown of LCN2 in PC-3 cells. LCN2 expression was analyzed 48 h after transfection of scrambled siRNA (scr siRNA) or indicated concentrations of siLCN2. (**A**) Western blot analysis showed strong reduction in LCN2 protein content (n = 4). GAPDH expression served as a control to demonstrate equal protein loading. (**B**) Quantitative mRNA analysis of *LCN2* revealed significant reduction in siLCN2 (35 nM) compared to scr siRNA (n = 3; *** *p* < 0.001). *LCN2* mRNA expression was normalized to *GAPDH* expression and given in relation to scr siRNA-treated cells. (**C**) siRNA-mediated knockdown of LCN2 is associated with reduced protein expression of fibronectin, IκBζ, Cx43 and IL-1β as determined by Western blot analysis (n ≥ 3). α-tubulin expression was used to demonstrate equal protein loading.

**Figure 3 cells-11-00260-f003:**
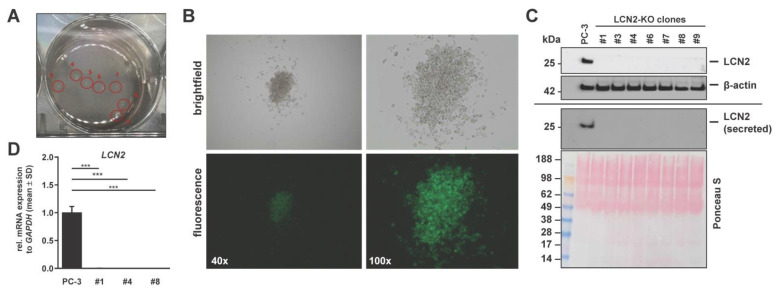
CRISPR/Cas9-mediated knockdown of LCN2 in PC-3 cells. LCN2-KO cells were seeded in low dilution in methylcellulose-based medium to establish single-cell-derived knockout cell lines. After being grown for 20 days in six-well plates, macroscopic (**A**) and microscopic (**B**) images of a representative clone of single-cell-derived LCN2-KO were taken. Bright-field and fluorescence images are shown in 40× and 100× magnification. (**C**) Western blot analysis verified knockout of LCN2 protein expression and secretion in all clones compared to parenteral PC-3 cells (n ≥ 3). β-actin served as an internal loading control for equal protein in lysates. Secreted LCN2 levels are shown with corresponding Ponceau S stain. Different cell clones are indexed by numbers. (**D**) Quantitative mRNA analysis showed successful *LCN2* depletion in LCN2-KO (#1, #4, #8) compared to PC-3 cells (n = 3). *LCN2* mRNA expression was normalized to *GAPDH* and given in relation to parenteral PC-3 cells. Differences between groups reaching significance are indicated as *** *p* < 0.001.

**Figure 4 cells-11-00260-f004:**
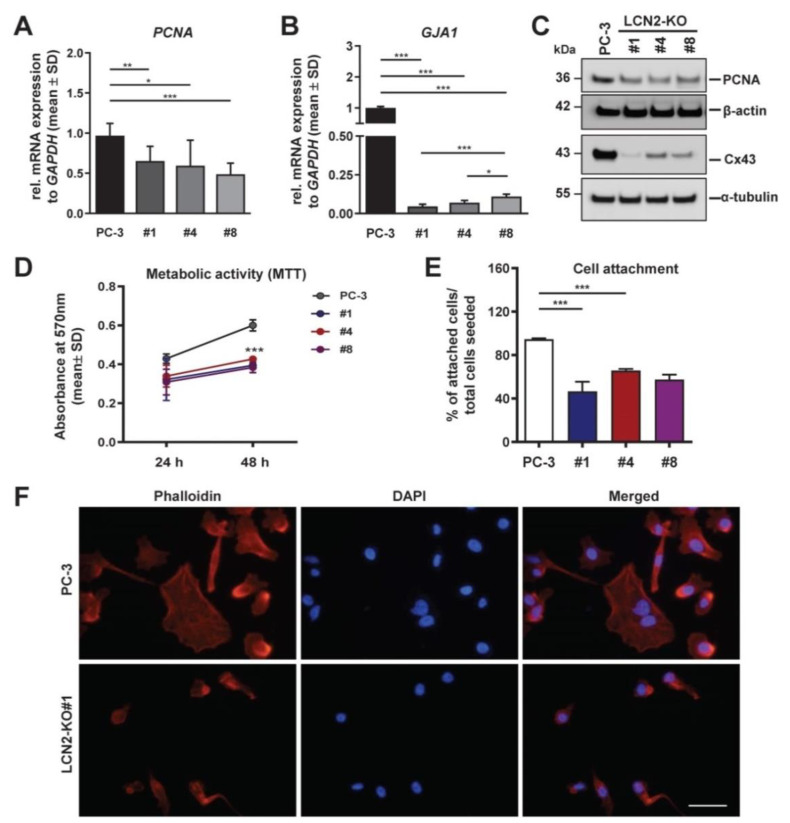
LCN2 deficiency in PC-3 cells results in changes in proliferation, adhesion, and cytoskeleton rearrangement. Quantitative mRNA expression analysis of (**A**) *PCNA* and (**B**) *GJA1* was performed in PC-3 and LCN2-KO (#1, #4, #8) cells grown in basal conditions (n = 3). Both markers were significantly reduced in all analyzed LCN2-KO cells compared to parenteral PC-3 cells. mRNA expression was normalized to *GAPDH* and are given in relation to parenteral PC-3 cells. Differences between the groups reaching significance are marked by asterisks (* *p* < 0.05, ** *p* < 0.01, *** *p* < 0.001). (**C**) Western blot analysis confirmed reduction in PCNA and Cx43 protein expression in LCN2-KO cells compared to PC-3 cells (n = 3). α-tubulin served as an internal loading control. (**D**) Metabolic activity was determined after 48 h in MTT assay (n = 3) and showed significantly reduced metabolic activity of all LCN2-KO cell lines (*** *p* < 0.001) compared to PC-3 cells. (**E**) Indirect cell attachment assay was performed as described in the Material and Methods section. Bars show the percentage of attached cells to total cells seeded (n = 3). All statistical values shown represent the comparison of *LCN2-*KO cell lines to the parenteral PC-3 cells. Differences between the groups reaching significance are indicated as *** *p* < 0.001. Note that all LCN2-deficient cell lines show significant reduction in cell attachment. (**F**) F-actin stress fibers were stained with Phalloidin-Rhodamine (red) in PC-3 and LCN2-KO#1 cells (n = 2). Nuclei were counterstained with DAPI (blue). Magnification: 400×, scale bar: 50 µm.

**Figure 5 cells-11-00260-f005:**
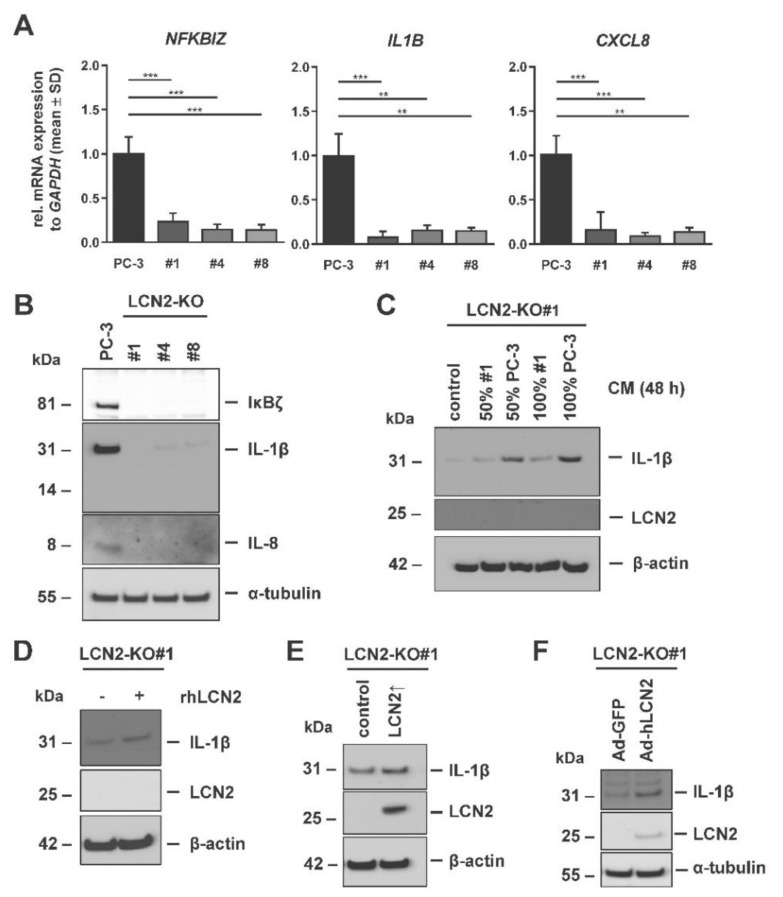
LCN2 depletion in PC-3 cells resulted in reduced expression of pro-inflammatory cytokines. (**A**) Quantitative mRNA expression analysis of *NFKBIZ*, *IL1B* and *CXCL8* was performed in PC-3 and LCN2-KO (#1, #4, #8) cells grown in basal conditions (n = 3). The mRNA expression of these genes was significantly reduced in all analyzed LCN2-KO cells test. mRNA expression was normalized to *GAPDH* and are given in relation to parenteral PC-3 cells. Differences between the groups reaching significance are indicated as ** *p* < 0.01, *** *p* < 0.001. (**B**) Western Blot analysis confirmed reduction in IκBζ, IL-1β and IL-8 protein expression in LCN2-KO cells compared to parental PC-3 cells (n = 3). (**C**) LCN2-KO#1 cells were stimulated for 48 h with indicated ratios of conditioned medium (CM) collected from PC-3 or LCN2-KO#1 cells. Western blot analysis revealed increased IL-1β quantities in LCN2-KO#1 cells, treated with CM from PC-3 cells (n = 3). (**D**) LCN2-KO#1 cells were stimulated with recombinant human LCN2 (rhLCN2, 100 ng/mL) for 24 h. Western blot analysis showed enhanced IL-1β expression in treated LCN2-KO#1 cells (n = 2). LCN2 was transiently reconstituted in LCN2-KO#1 cells by transfection with a (**E**) hLCN2 construct (n = 3) or by (**F**) adenoviral infection with Ad5-CMV-hLCN2 (Ad-hLCN2) (n = 2). Either a corresponding empty vector (control) or Ad5-CMV-GFP (Ad-GFP) served as controls. Cells were harvested for protein analysis after 24 h (**E**) or 48 h (**F**). Re-expression of LCN2 in LCN2-KO#1 cells resulted in increased IL-1β quantities. α-tubulin or β actin expression served as internal loading controls in Western blot analysis.

**Figure 6 cells-11-00260-f006:**
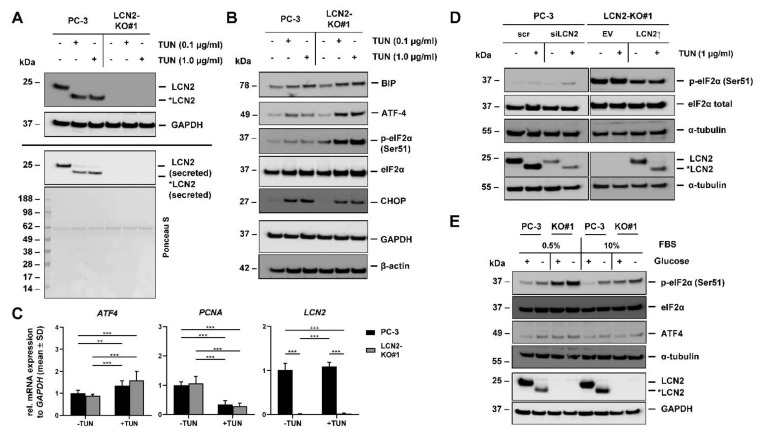
LCN2 deficiency affects response to ER stress in vitro. Parental PC-3 and LCN2-KO#1 cells were treated with indicated concentrations of tunicamycin (TUN) for 24 h. Supernatants were collected, and lysates harvested for protein or mRNA analysis. (**A**) TUN treatment prevents glycosylation of LCN2 in PC-3 cells as evidenced by detection of non-glycosylated form of LCN2 (*LCN2) in both cell lysates and supernatants. Secreted LCN2 levels are shown with corresponding Ponceau S stain. (**B**) TUN treatment enhanced different unfolded protein response (UPR) markers on protein level (n ≥ 3) in LCN2-KO#1 and PC-3 cells. (**C**) Quantitative mRNA analysis of *ATF4*, *PCNA* and *LCN2* was performed in PC-3 and LCN2-KO#1 cells treated with TUN (1 µg/mL) for 24 h (n = 3). Significant changes were detected between LCN2-KO cells compared to parenteral PC-3 cells. mRNA expression was normalized to *GAPDH* and given in relation to expression in parenteral PC-3 cells. Differences between the groups reaching significance are indicated as a ** *p* < 0.01, and *** *p* < 0.001, respectively. (**D**) PC-3 cells were treated with siLCN2 (35 nM) or scr siRNA for 24 h, while expression in LCN2-KO#1 cells were restored by transfection of a plasmid directing hLCN2 expression. Cells transfected with empty vector (EV) served as control. After 24 h, both cell lines were treated with TUN (1 µg/mL) for further 24 h and Western blot analysis were performed to detect p-eIF2α and LCN2 (n ≥ 2). (**E**) PC-3 and LCN2-KO#1 cells were cultured in medium with or without glucose (containing 0.5% or 10% FBS) for 48 h. Western blot analysis showed induction of p-eIF2α and ATF-4 in both cell lines, as well as non-glycosylated form of LCN2 (*LCN2) in PC-3 cells under glucose deprivation conditions (n = 3). In Western blot analysis, α-tubulin, β-actin, or GAPDH expression served as internal protein loading controls.

## Data Availability

The original data of this study is stored in the Institute of Molecular Pathobiochemistry, Experimental Gene Therapy and Clinical Chemistry (IFMPEGKC) located at the RWTH University Hospital Aachen. Results of repetitions that are not show here can be requested from the corresponding author.

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
