# Peer review of "Lipocalin-2 (LCN2) Deficiency Leads to Cellular Changes in Highly Metastatic Human Prostate Cancer Cell Line PC-3"

_cells, 2022, doi:10.3390/cells11020260_

Round 1

Reviewer 1 Report

In this manuscript, Sarah K. Schroder et al reported LCN2 is related to the metastatic characteristics in PC3 cells by molecular analysis of IL-1beta, Cx43, inflammatory cytokines, adhesion markers, F-actin, p-eIF2alpha as well as phenotypic evaluation upon LCN2 knockdown or knockout including endoplasmic stress response and unfolded protein response. The molecular characterization is extensive and the discussion over the phenotypic changes are relevant to the results. However, the major concern I have is the relative disconnection from the title to the results/discussion. The title implies LCN2’s pro-metastatic role in prostate cancer, though it’s only confined to one specific cell line PC-3 which is metastatic as introduced the result section. However, most of the molecular changes and phenotypic changes are not closely related to metastasis. For the least, an in-vivo animal experiment like mouse tail vein injection model or more strictly speaking, an orthotopic prostate cancer mouse model is required to validate the LCN2’s involvement in metastasis.

Author Response

Dear reviewer,

please find our response to your comments in the attached pdf-File.

Reviewer 2 Report

In this manuscript Schröder et al investigate the role of Lipocalin 2 (LCN2) in prostate cancer. By using siRNA or CRISPR/Cas9-mediated knockout cell lines, the authors report reduced LCN2 is accompanied by decreased expression of IL-1β and Cx43. In addition, reduced proliferation, lower adhesion and disrupted F-actin distribution are observed in LCN2 deficient PC-3 cells. Furthermore, LCN2-deficient PC-3 Cells are prone to ER stress.

The oncogenic effect of LCN2 has been proposed and implicated in a variety of cancer types. LCN2 gene has been shown to promotes breast cancer progression by inducing EMT through the ERα/Slug axis. (Yang et al., PNAS 2009,106:3913-8). In prostate cancers, it is reported that LCN2 promotes cell migration and invasion by inducing EMT through the ERK/SLUG axis. (Ding et al., Prostate 2015,75:957).

In this manuscript, data are well presented and methodologies are solid. However the significance and novelty of this study is low. More importantly, only phenotypes are examined in LCN2-deficient cells, related mechanism is not investigated. Therefore,

insights are not provided to help better understand the oncogenic role of LCN2 in prostate cancers. Additional experiments are highly suggested:

  1. The study reports that LCN2-deficient PC-3 cells show reduced proliferation, adhesion and disrupted F-Actin Stress Fibers, which are not surprising since LCN2 induces EMT through ERK/Slug axis in prostate cancer or ERα/Slug axis in breast cancer. It is more important to ask how does LCN2 activates ERK signaling? Since PC-3 also expresses ERα, whether EMT induced by LCN2 involves ERα in prostate cancer too? What proteins LCN2 directly binds to induce EMT?
  2. The study presents results that depletion of LCN2 in PC-3 cells results in higher levels of p-eIF2α and increased UPR upon cell stress, suggesting a role of LCN2 protecting PC-3 from being overwhelmed by UPR. Again, how does LCN2 affect the expressions of UPR markers? Is it the cellular or secreted LCN2 that plays the role? The relevant mechanism remains largely unknown.

In sum, this study presents phenotypes from several aspects in LCN2-deficient cells. However, similar results have been previously reported in prostate cancer or other cancer types. The study fails to reveal deeper mechanistic insights on the phenotypes observed. Additional experiments deciphering mechanisms are highly suggested before the acceptance of the manuscript is considered.

Author Response

(The authors gave the same response as above.)

Reviewer 3 Report

Congratulations to the authors for nicely exploring the putative role of LCN2 in prostate cancer.

They also perform a deep literature review about previous publications that already suggest their findings. So that, the specific novelty of their research is only exploring some associations and events such as cell-cell communications and inflammations, but the authors itself comment on the need of further explore the mechanisms suggested.

There are some concerns that should be address:

  • First, the authors only explore one specific cell line that highly express LCN2 and downregulate it to explore different hypothesis. It would be of great interest to explore other cell lines, also with lower expression and trying to overexpress it.
  • It is known that cell line environment is very specific so extrapolating the results shown from only one cell line to patients is quite a big challenge. It is necessary to at least further explore the authors hypothesis and findings with an “in vivo” approach. Also, as free available, all the association suggested should be checked in public data set such as GRASSO, TCGA, MSKK…
  • When the authors explore the association of LCN2 and IL-B, also comment about its role in LNCaP aggressiveness. Why do not the authors further explore this hypothesis in their LNCaP cells?

Author Response

(The authors gave the same response as above.)

Round 2

Reviewer 1 Report

In the revised manuscript, the authors revised the title in accordance with the results and provides better data interpretation. But due to the lack of in-vivo results and the limitation of cell line bias, the translational significance of this study is greatly compromised as to how much extent LCN2 might also work similary in other cell lines in PCa or whether this protein deregulation induces phenotypic changes.

Reviewer 2 Report

The revised version of the manuscript together with authors' response reflect the thorough thinking of the investigators on the experiments design and scientific questions they trying to answer. They've addressed the questions to some extent the reviewer asked, and proposed a mechanism for the phenotypes. Therefore, the reviewer would agree to accept the manuscript in present form.

Reviewer 3 Report

Congratulations to the authors for the revision of the paper.

All the concerns have been adequately responded.